# Projection of North Atlantic Oscillation and its effect on tracer transport

Sara Bacer[1], Theodoros Christoudias[2], and Andrea Pozzer[1]

[1]Atmospheric Chemistry Department, Max Planck Institute for Chemistry, Mainz, Germany
[2]Computation-based Science and Technology Research Center, The Cyprus Institute, Nicosia, Cyprus

*Correspondence to:* S. Bacer (sara.bacer@mpic.de)

**Abstract.** The North Atlantic Oscillation (NAO) plays an important role in the climate variability of the Northern Hemisphere with significant consequences on long-range pollutant transport. We investigate the evolution of pollutant transport in the 21st century influenced by the NAO under a global climate change scenario. We use a free-running simulation performed by the ECHAM/MESSy Atmospheric Chemistry (EMAC) model coupled with the ocean general circulation model MPIOM, covering the period from 1950 until 2100. Similarly to other works, the model shows a future northeastward shift of the NAO centres of action and a weak positive trend of the NAO index (over 150 years). Moreover, we find that NAO trends (over periods shorter than 30 years) will continue to interchange in the future. To investigate the NAO effects on transport we consider carbon monoxide tracers with exponential decay and constant interannual emissions. We find that at end of the century during positive NAO phases south-west Mediterranean and Africa will see higher pollutant concentrations with respect to the past, while a wider part of north Europe will benefit from increased pollutant depletion. Such results are confirmed by the changes observed in the future for tracer concentration and vertically integrated tracer transport, differentiating the cases of "high NAO" and "low NAO" events. Therefore, under a global climate change scenario, the local air quality conditions over Europe and North Africa, influenced by North Atlantic teleconnection activity, will become more extreme.

## 1  Introduction

The North Atlantic Oscillation (NAO) is the most prominent recurrent pattern of atmospheric variability over middle and high latitudes in the Northern Hemisphere (NH). It is a swing between two pressure systems, the Azores High and Icelandic Low, which redistributes atmospheric masses between the Arctic and the subtropical Atlantic influencing weather conditions (Walker et al., 1932). When the Icelandic Low and Azores High are relatively stronger, the pressure difference is higher than average (positive NAO phase) and the north-south pressure gradient produces surface westerlies stronger than average across the middle latitudes of the Atlantic towards northern Europe. On the other hand, when the low and high surface pressure are relatively weaker (negative NAO phase) the flow has a reduced zonal component. These meridional oscillations produce large changes in the mean wind speed and direction, heat and moisture transport, surface temperature and intensity of precipitation, especially during boreal winter (Hurrell et al., 2003, and references therein). Several studies (Hurrell, 1995; Visbeck et al.,

2001; Hurrell et al., 2003) have associated the westerly flow during positive NAO with warm and moist maritime air and enhanced precipitation over northern-western Europe, and colder and drier conditions over the Mediterranean.

As the NAO exerts a strong influence on the boreal winter weather, it can also affect the transport of gas pollutants on hemispheric scale. Li et al. (2002) examined the transatlantic transport of anthropogenic ozone and the NAO impacts on the surface ozone in North America and Europe; they found that there are higher northern American ozone concentrations at Mace Head Ireland during positive NAO, when westerly winds across the North Atlantic are stronger. Creilson et al. (2003) also analysed the relationship between the NAO phases and the tropospheric ozone transport across the North Atlantic and discovered that rises of ozone over western Europe are strongly correlated with positive NAO. Eckhardt et al. (2003) studied the relationship between the NAO and transport towards the Arctic and found that concentrations of surface carbon monoxide, originating both from Europe and North America, increase in the Arctic during the NAO positive phases. Christoudias et al. (2012) studied the transport of regionally-tagged idealised tracers in relation to the NAO and found that, during high positive NAO phases, the trace gases emitted from North America are transported relatively far to northern-eastern Europe, while the trace gases emitted over Europe are transported mostly over Africa and the Arctic Circle. Pausata et al. (2012) showed both with station measurements and coupled atmosphere-chemistry model simulations that the NAO affects surface ozone concentrations during all seasons, except in autumn. The sensitivity studies by Thomas et al. (2014) regarding the free tropospheric carbon monoxide concentrations to different atmospheric weather conditions confirmed the NAO control of pollutant distribution and transport over the region of Nordic countries.

A number of studies have focused on the impacts of the NAO on aerosol concentrations. Moulin et al. (1997) analysed the role of the NAO in controlling the desert dust transport into the Atlantic and Mediterranean and suggested that the NAO likely influences the distribution of anthropogenic aerosols. Jerez et al. (2013) investigated the NAO influence on European aerosol concentrations through local atmospheric processes (e.g., precipitation, wind, cloudiness) and found that positive NAO promotes higher ground-level aerosol concentrations in southern regions of the Mediterranean during winter. Pausata et al. (2013) proved the influence of the NAO extreme events during the 1990s on the variability of particulate matter concentrations over Europe and suggested the usage of the NAO index as proxy for health impacts of pollution.

The aforementioned studies suggest that future NAO phases will be important when projecting the northern American and European pollutant transport over Europe and the Arctic. There are, however, no studies on the influence of the NAO on tracer and pollutant transport under a future scenario, taking into account the global climate change. With this work we aim to study the NAO influence on the pollutant transport in the future.

The NAO is an intrinsic mode of atmospheric variability but there is mounting evidence in the literature that it is unlikely that only stochastic atmospheric processes are the cause of NAO changes. There are a few candidate mechanisms to interpret low-frequency variations such as the North Atlantic (Rodwell et al., 1999) and tropical (Hoerling et al., 2001) sea surface temperature (SST), the sea-ice variations in the North Atlantic Ocean (Mehta et al., 2000) and the stratospheric circulation (Baldwin et al., 2001). Recently, Woollings et al. (2015) have ascribed the NAO variability on interannual-decadal timescales to the latitudinal variations of the North Atlantic jet and storm track, and the NAO variability on longer timescales to their speed and strength changes. In order to explain the upward trend observed from 1960s until 1990s some external forcings have

been proposed as responsible. They include the increase of greenhouse gases (Kuzmina et al., 2005), warmer tropical SST (Hoerling et al., 2001) and the strengthened stratospheric vortex (Baldwin et al., 2001). However, there is still no consensus and Osborn et al. (1997) asserted that recent variations can not be explained, even when combining the anthropogenic forcing and internal variability. Thus, a conclusive understanding of past NAO variability has still to be reached and the future NAO evolution continues to be an open research topic.

Earth system model simulations with increasing greenhouse-gas (GHG) concentrations can provide projections of the NAO and future trends. Most models have projected a weak positive NAO trend under a global warming climate change scenario. Gillett et al. (2013) found this when considering the mean of 37 CMIP5 models' merged historical and RCP 4.5 simulations for each season, and Stephenson et al. (2006) obtained similar results with 14 models out of 18 studied. However, some studies found the NAO index in a future scenario only weakly sensitive to the GHG increment, with no significant trends (Fyfe et al., 1999; Dorn et al., 2003; Rauthe et al., 2004; Fischer et al., 2009), or even decreasing trends (Osborn et al., 1997). More lately, Pausata et al. (2015) analysed the impacts due to the aerosol reduction (after air pollution mitigation strategies) and GHG increment on the winter North Atlantic atmospheric circulation and obtained a stronger positive NAO mean state by 2030. The dependency of the results on the model used is still unclear (Gillett et al., 2003; Stephenson et al., 2006). Other research questions are still open, regarding which climate processes govern the NAO variability, how the phenomenon varies in time, and what is the potential for the NAO predictability (Visbeck et al., 2001; Hurrell et al., 2001; Woollings et al., 2015).

The distribution and development of gases and aerosols are controlled by atmospheric chemistry and physics, including the transport of air masses integrated over continental scale. A large number of studies have addressed the NAO influence on tracer transport and the future trends of the NAO as disparate topics. However, no study has addressed the NAO influence on the pollutant transport throughout the 21st century using an integrated modelling approach and with a full atmospheric chemistry to account for all potential feedbacks.

The aim of this paper is to study the influence on the pollutant transport due to the NAO in the span of the 21st century using a full Earth system model. We analyse a simulation performed by a coupled atmosphere-chemistry-ocean general circulation model in order to *(i)* investigate the NAO signal and trend in the future and *(ii)* study the NAO influence on the pollutant transport in the past and in the future over the North Atlantic sector. For the analysis, we focus on the carbon monoxide (CO) pollutant, which is directly emitted by combustion sources and has a lifetime of 1–3 months in the atmosphere; thus, it has a sufficiently long atmospheric residence lifetime relative to the timescales of transport.

The paper is structured as follows: Sect. 2 briefly describes the model used and the simulation set-up; Sect. 3 presents the NAO trends of the future projection; Sect. 4 analyses the NAO influence on and the changes of tracer transport. Conclusions and outlook are given in Sect. 5.

## 2  Methodology

Increasingly, the dynamics and chemistry of the atmosphere are being studied and modelled in unison in global models. Starting with the fifth round of the Coupled Model Intercomparison Project Phase 5 (CMIP5), some of the Earth system

models (ESMs) that participated with interactive oceans included calculations of interactive chemistry. It was also a main recommendation of the SPARC CCMVal Report (2010), that Chemistry-Climate Models (CCMs) should continue to evolve towards more comprehensive, self-consistent stratosphere-troposphere CCMs. These developments allow for including a better representation of stratosphere-troposphere, chemistry-climate and atmosphere-ocean couplings in CCMs and ESMs used for
more robust predictions of climate changes and mutual influences and feedbacks on emitted pollutants (Eyring et al., 2012). The EMAC model was one of the first community models to introduce all these processes (Jöckel et al., 2006).

In this work we analyse a long chemistry climatic simulation performed by the ECHAM/MESSy Atmospheric Chemistry (EMAC) climate model under the "Earth System Chemistry Integrated Modelling" (ESCiMo) initiative (Jöckel et al., 2016). The EMAC model is a numerical chemistry and climate simulation system which uses the Modular Earth Submodel Sys-
tem (MESSy) to describe tropospheric and middle atmosphere processes and their interactions with oceans, land and human influences via dedicated sub-models (Jöckel et al., 2010).

The long chemistry climatic simulation RC2-oce-01 (Jöckel et al., 2016), hereafter referred to as "coupled simulation", simulates the development of the climate covering the period 1950–2100. The simulation is performed by the fully coupled atmosphere-chemistry-ocean model EMAC-MPIOM (Pozzer et al., 2011), using the 5th generation European Centre Hamburg
general circulation model (ECHAM5, Roeckner et al. (2006)) as the dynamical core of the atmospheric model and the MESSy submodel MPIOM (Max Planck Institute Ocean Model, Marsland et al. (2003)) as the dynamical core of the ocean model. The simulation resolution uses a spherical truncation of T42 (corresponding to a quadratic Gaussian grid of approx. $2.8 \times 2.8$ degrees in latitude and longitude) and 47 vertical hybrid pressure levels up to $0.01$ hPa into the middle atmosphere (approximately $80$ km with a resolution of $\sim 20$ hPa ($\sim 1$ km) at the tropopause), referred to as T42L47MA. This vertical resolution is
essential in order to take into account the influence of the stratosphere on the NAO variability (Baldwin et al., 2001). Proper representation of the stratospheric dynamics is important for simulating future climate changes and a realistic reproduction of the NAO changes (Shindell et al., 1999). Scaife et al. (2007) further showed that the stratospheric variability has to be reproduced in order for models to fully simulate surface climate variations in the North Atlantic sector. The resolution for the ocean corresponds to an average horizontal grid spacing of $3 \times 3$ degrees with 40 unevenly spaced vertical levels (GR30L40).
An important feature of the EMAC model is its capability to provide a careful treatment of chemical processes and dynamical feedbacks through radiation (Dietmüller et al., 2016). Thus, the coupled simulation includes gas phase species computed online trough the MECCA submodel (Sander et al., 2011), while it uses only a climatology of atmospheric aerosols to take into account the interactions with radiation and heterogeneous chemistry. The model incorporates anthropogenic emissions as a combination of the ACCMIP (Lamarque et al., 2013) and RCP 6.0 scenario (Fujino et al., 2006). A detailed description can be
found in Jöckel et al. (2016) and references therein. Let us stress that the same EMAC model forced with SST has been already used by Christoudias et al. (2012) to successfully reproduce the NAO.

Coupled general circulation models (GCMs) perform better than atmospheric GCM forced with SST in reproducing the spatial patterns of atmospheric low-variability and the NAO phenomenon (Saravan, 2011). Several works have shown that coupled models are able to simulate the main features of the NAO (e.g., Osborn et al. (1997), Stephenson et al. (2006)). Re-
cently, Xin et al. (2015) have quantified the contribution of the coupling in the NAO variability, showing that 20% of the

NAO monthly variability is caused by the ocean-atmosphere coupling and $80\%$ is due to the chaotic atmospheric variability. Therefore, a coupled model is essential for a reasonable projection of future NAO. Our model is one of the first to include a full ocean-atmosphere coupling, stratospheric circulation in conjunction with online chemistry and anthropogenic emissions to capture all associated feedbacks, thus providing state-of-the-art simulation capability of the phenomenon and potential impacts.

In order to investigate the transport of pollutants we use passive tracers with emissions modelled after CO emissions for the year 2000 (i.e. no interannual variability) and decay lifetime constant in time. These tracers are well-suited for investigating transport-related effects as no chemical influences or emission variability are included. CO is a good proxy for anthropogenic pollution, as it is mostly emitted by biomass burning and human activities (Pozzer et al., 2007). In particular, we consider two passive CO tracers with a constant exponential decay (e-folding time) equal to 25 and 50 days, referred to as CO_25 and

CO_50 respectively. For the whole analysis we focus on the winter (DJF: December-January-February) seasonal means, since the sea level pressure (SLP) amplitude anomalies are larger in winter and the NAO is typically stronger in this period. To study the intercontinental transport of CO (Subsect. 4.2) we compute the vertically integrated tracer transport vector, defined as:

$$\boldsymbol{Q} = \frac{1}{g} \int\limits_{0}^{ps} r\boldsymbol{u}dp, \tag{1}$$

where $r$ is the mixing ratio of the tracer (i.e. CO, CO_25 or CO_50) in $mol/mol$, $\boldsymbol{u}$ the horizontal wind speed, $p$ the atmo-

15 spheric pressure, $ps$ the surface pressure and $g$ the gravitational acceleration.

## 3   NAO representation and changes

### 3.1   NAO representation

In order to define the spatial structure and temporal evolution of the NAO we use Empirical Orthogonal Function (EOF) analysis. We compute the eigenvectors of the cross-covariance matrix of the time variations of the SLP (Hurrell et al., 2003).

By definition the eigenvectors are spatially and temporally mutually orthogonal and scale according to the amount of the total variance they explain; the leading EOF (EOF1) explains the largest percentage of the temporal variance in the dataset. The NAO is identified by the EOF1 of the cross-covariance matrix computed from the SLP anomalies in the North Atlantic sector. The EOF1 spatial pattern is associated with a north-south pressure dipole with its centres of action corresponding to the NAO poles with highest SLP variability. Therefore, we compute the EOF1 from winter seasonal SLP anomalies in the North Atlantic

sector (20°N-80°N, 90°W-40°E) and we find that the long chemistry coupled simulation (1950-2100) reproduces the NAO signal with the typical north-south dipole structure (Fig. 1, top). The EOF1 explains $38.8\%$ of the total variance, in accordance with the results found in literature (e.g., Fischer et al. (2009), Ulbrich et al. (1999)). In order to detect the NAO differences between the past and the end of the 21st century, we define two 30-years-long periods referred to as "recent past" (1980-2010) and "future" (2070-2100). Fig. 1 (centre and bottom) shows the EOF1 analysis for the two distinct periods. A climatological

timescale (30 years) for the two periods has been chosen to reduce the interdecadal variability. Additionally, we have chosen various climatological timescales of 30 years during the past and future and we have computed the EOF1 in all periods, i.e.

1950-1979, 1960-1989, 1970-1999, 1980-2009 in the past and 2040-2069, 2050-2079, 2060-2089, 2070-2099 in the future. The results (shown in the electronic supplement) exibit differences between the two climatological periods, but they do not between any of the decadal timescale within each period. Thus, the changes observed between past and future NAO patterns are not due to decadal variability but rather they are climate induced.

In Fig. 1 we can see that the centres of action of the NAO move northeastward towards the end of the century. Such NAO shift is in agreement with the results obtained by Ulbrich et al. (1999), Hu et al. (2003) and Pausata et al. (2015) for a climate change global warming scenario.

    The shift of the NAO centres of action has to be taken into account when examining the temporal evolution of the NAO pattern. The NAO station-based index, defined as the difference of the normalized SLP between one northern station in Iceland

and one southern station in the Azores, is fixed in space and is not able to capture the spatial variability of the NAO centres of action over seasonal (Hurrell et al., 2003) or (future) decadal (Ulbrich et al., 1999) scales. Since our model projects a spatial shift of the NAO centres, we will be considering the principal component time series of the leading EOF of SLP (PC1) (Hurrell et al., 2003) as NAO temporal index. The PC1 computed for the entire simulation (1950-2100) after subtracting the SLP climatology of 1980-2010 is shown in Fig. 2.

**3.2   NAO changes**

To investigate the NAO temporal variability and trends we calculate the linear regression coefficients of the PC1 computed for the entire 150-years simulation (Fig. 2) considering sliding windows. In particular, we define windows of variable length between a minimum of 10 years and a maximum equal to 150 years sliding along the whole PC1 time series. We compute the linear slope (trend) for each window and assign the value to the window central year (e.g., the regression coefficient of the PC1

series in the selected period 1980-1990, a 11-year window, is assigned to the year 1985). Results in Fig. 3 show that no change in the projected future NAO variability is identified compared to the past when considering periods shorter than 30 years. For windows of length between 30 and 60 years, upward trends (centred in the 1980s and 2040s) interchange with downward trends (centred in the 2010s and 2060s). On longer window lengths we find that very weak non-statistically significant NAO trends are prevalent. The slope of the overall trend computed for the entire PC1 is $2.99 \times 10^{-3} \pm 0.95 \times 10^{-3}$ (i.e. significant

at $95\%$). In summary, our coupled EMAC-MPIOM model predicts a small significant positive trend for the NAO (for the 150 years horizon) in agreement with other studies that have used coupled models (e.g., Gillett et al. (2003), Hu et al. (2003), Stephenson et al. (2006)). In the same plot (Fig. 3), we have marked two triangles in correspondence to the recent past and future periods, with the aim of stressing the NAO trend changes. In the lower triangle we distinguish two sharp patterns: an upward trend (red shading) which dominates between 1980 and 1991 and a downward trend (blue shading) which dominates

from 1992 onwards. Differently, in the upper triangle we note that, at the end of the century, there is a clear prevalence of positive NAO trends.

    To enhance the analysis of NAO temporal evolution, we compute the number of (winter) NAO phases over 30 years, in the recent past and in the future (Fig. 4). In such a way, we study how the distribution of NAO phases evolves. In the recent past (Fig. 4, left) the distribution covers a large PC1 interval ($[-3; 2.5]$) and the number of NAO phases is at most 3, except in the

interval $[0; 1]$ where it is clearly higher (equal to 9 and 10). Differently, in the future (Fig. 4, right) the distribution is skewed towards positive values of PC1 (the interval is $[−1.5; 2.5]$), with numbers of NAO phases between 4 and 7 in the interval $[−1.5; 1]$ and between 1 and 3 in the interval $[1; 2.5]$. Therefore, at the end of the century the number of negative NAO phases increases (15 in the future vs. 10 in the past), vice-versa, the number of positive NAO phases decreases (16 in the future vs. 21 in the past). However, the "high NAO extreme" events (PC1 $> 1.5$) are more frequent in the future (4 in the future vs. 1 in the past), while the number of "low NAO extreme" events (PC1 $< −1.5$) decreases (0 in the future vs. 3 in the past).

## 4 NAO effects on tracer transport in the future

### 4.1 Correlation and regression analysis

In order to investigate the NAO influence on tracer transport we compute the correlation (Fig. 5) and the regression (Fig. 6) between the PC1 and tracer mixing ratio at the surface level. We consider passive tracers whose emissions and decay lifetime are constant (CO_25 and CO_50) in order to remove influences by chemical production/decomposition variability. In this way the analysis gives information purely on the effect of tracer transport. We perform the correlation and regression considering the CO_25 tracer, which undergoes exponential decay with e-folding time equal to 25 days. A supplementary analysis is repeated for CO_50, with 50-days e-folding constant, to provide a constraint on the systematic uncertainty associated with the resident time of the tracer in the atmosphere and to show robustness of our results. To identify the future changes in transport pathways related to the NAO we perform the analysis separately in the two periods, recent past and future.

By means of the correlation (Fig. 5) we determine where European and Eastern USA CO-like tracers have a linear relationship with NAO. Higher the correlation (in absolute value), stronger the linear dependence between tracer mixing ratio and PC1. In particular, a positive correlation implies that tracer concentrations are higher (or lower) with an increase (or decrease) of the PC1, while a negative correlation implies that concentrations are lower (or higher) with an increase (or decrease) of the PC1. The intensity of such relationship will be further discussed later with the regression calculations (Fig. 6). We observe that, in the recent past (Fig. 5, left), the PC1 and CO_25 mixing ratio are significantly correlated over the northern part of the Northern America east coast, north-west Baffin Bay region, Arctic, north Africa and part of the Iberian Peninsula. Also present is a continuous area of significant anti-correlation encompassing the American central-east coast (near Florida and Cuba), through the central North Atlantic Ocean, towards northern and eastern Europe, and the Black Sea regions. The analysis with CO_50 leads to similar results (see the electronic supplement) and, thus, the outcomes can be considered robust under the uncertainties associated with pollutant tracer atmospheric residence lifetimes.

Since the CO concentration over Europe is mostly influenced by emissions from Europe and only partially from North America (the Asian contribution can be considered negligible (Duncan et al., 2008)), we can compare our results with the findings of Christoudias et al. (2012) that used tracers tagged by origin. We find transport patterns (Fig. 5, left) similar with the ones of Christoudias et al. (2012) for European emissions. However, our results supersede those in Christoudias et al. (2012) as that study was limited in the period 1960–2010 and was forced by prescribed SST and global atmospheric hydroxyl radical (OH) concentrations (as the removal mechanism for CO depletion).

The analysis regarding the future period is displayed in Fig. 5 (right). All significantly correlated areas increase in size compared to the past, except for the area at north-west of the Baffin Bay which decreases. The area with positive correlation over the Arctic spreads southwards up to the Scandinavian Peninsula and the one over Africa spreads westwards and northwards, covering further the Iberian Peninsula. Moreover, the correlation over north-west Africa and the near ocean becomes stronger
with values between 0.6 and 1.0, greater than in the past. Similarly, the area with significant anti-correlation is wider with respect to the past, and a more pronounced gradient between positive and negative correlated regions is formed. The large area with negative correlation over the North Atlantic extends further over northern Europe and the North Atlantic Ocean and develops over new regions, i.e. south Greenland and Baffin Bay. The magnitude of the negative correlation also increases over north-east Europe, south Scandinavian, and the North Atlantic Ocean (between Great Britain and Iceland) with values
in the range $-0.6$ and $-0.8$. Again, the analysis considering CO_50 has produced similar results (presented in the electronic supplement).

In order to better define the relationship between NAO and tracer transport, we compute the regression between CO_25 mixing ratio and PC1 (Fig. 6). Here, analogously to the correlation, areas with positive values mean that positive/negative NAO phases drive a higher/lower stagnation of trace pollutants, while areas with negative values mean that positive/negative
NAO phases drive a depletion/increment of such pollutants. However, differently from the correlation, the regression map shows how intense the effect of NAO on CO_25 concentration could be. We observe that patterns in Fig. 6 are very similar to those in Fig. 5. The regression analysis shows that the flow over Europe transports tracers over the Arctic, south Mediterranean, and Africa during positive NAO phases and splits the European continent in two distinct areas. Conversely, during negative NAO phases, the air is more stagnant over Central Europe allowing pollutants to accumulate. Such results extend what have
been found by Creilson et al. (2003), Eckhardt et al. (2003) and Christoudias et al. (2012), who analysed the NAO effects on ozone, carbon monoxide and origin-tagged idealised tracers, respectively. Comparing the recent past and future periods, we find that this dichotomy over Europe is further stressed in the future, which is mostly characterized by positive NAO trends (Fig. 3) and more high NAO extreme events (Fig. 4) with respect to the past. Indeed, a stronger Azores High (during positive NAO) leads to enhanced transatlantic transport towards north-east Europe and then southwards, over Africa, and to a stronger
separation of the flow over Europe. Therefore, northward transport to the Arctic and southward transport to Africa are further enhanced of in the future. Such considerations are confirmed when studying the differences between future and recent past tracer concentration and transport (see next subsection).

Consequently, at the end of the century during positive NAO phases south-west Mediterranean and north Africa will suffer from higher pollutant concentrations, while a wider part of north Europe will benefit from lower concentrations of long range
pollutants, associated with improved surface air quality. Similarly, the splitting over the American east coast will be enhanced as well, to a lesser degree. Nevertheless, we should note that this work is related only on the transport of CO-like tracers with constant lifetime and emissions, and thus it does not account for a possible (and probable) decrease of pollutant emissions both over Northern America and in Europe. Moreover, we do not deal with the reduction of aerosol and aerosol precursors emissions, predicted by most of the Representative Concentration Pathways (RCP, Lamarque et al. (2013)) over the Mediterranean, since
we focus on trace gases rather than aerosols.

## 4.2 Tracer concentration and transport changes

Here, we further develop our analysis differentiating high and low NAO events, both in the recent past and in the future. We define "high NAO" and "low NAO" as (winter) periods with PC1 higher than $0.5$ and lower than $-0.5$, respectively. We obtain 12 high and 8 low NAO phases in the recent past and 9 high and 11 low NAO phases in the future (therefore averages are always computed over at least 8 values).

Firstly, we compute the temporal averages of CO_25 winter surface mixing ratio during high and low NAO events in order to investigate how tracer concentration changes in the future. In Fig. 7 we show the differences (percentages) between future and recent past during high and low NAO periods. We observe that in the future, during high NAO (Fig. 7, left), concentrations increase by $10\%$ over north Africa and Mediterranean and even by $15\%$ over some areas of the Iberian Peninsula, Greece and Aegean Sea; concentrations are lower than in the past over northern Europe and Greenland (in the range down to $-10\%$). On the other hand, during low NAO (Fig. 7, right) CO_25 concentrations increase over north-east Africa and west-centre Europe (up to $15\%$) and decrease over north Scandinavian, the Arctic, and some areas of North America and Atlantic Ocean (down to $-10\%$). Therefore, we corroborate the results of the previous subsection and, moreover, we estimate which concentration changes are associated to the different NAO phases. Nevertheless, we would like to stress that, while the correlation and regression analyses (Fig. 5 and Fig. 6) were performed over 30 years, here the years considered are fewer (having to satisfy the conditions PC1 $< -0.5$ or PC1 $> 0.5$).

Secondly, following the same definitions of high and low NAO, we compute the temporal averages of $Q$ (1). The main features of transport are in agreement with Hurrell (1995) and Christoudias et al. (2012): during positive NAO (Fig. 8, top) the axis of maximum transport has a southwest-to-northeast orientation across the Atlantic and extends farther to north-east Europe, while during negative NAO (Fig. 8, bottom) it is more longitudinally oriented. Comparing the recent past and future, we observe that during high NAO (Fig. 8, top) the east-northward transport of CO_25 is more pronounced in the future over the North Atlantic Ocean, from the Northern America coast towards Ireland, while it gets weaker over south Greenland, Mediterranean and west Europe. During low NAO (Fig. 8, bottom), the eastward CO_25 transport over the North Atlantic Ocean extends farther eastwards in the future, while it decreases over the Mediterranean Sea; differently from the high NAO case, transport gets slightly stronger over south Greenland in the future. The main future changes of CO_25 transport, which gets generally stronger over the North Atlantic Ocean and weaker over the Mediterranean, confirm information retrieved from the correlation and regression analysis.

## 5 Conclusions

A free-running simulation performed by the coupled EMAC-MPIOM model has been analysed in order to study the influence of the NAO on future pollutant transport and concentration changes. The simulation takes into account the GHG increment during the 21st century according to the ACCMIP (Lamarque et al., 2013) and RCP 6.0 scenario (Fujino et al., 2006) and uses an atmospheric aerosol climatology. The model is able to reproduce the SLP anomalies and the NAO signal (Christoudias et al.,

2012), and the EOF analysis performed with the coupled simulation shows the typical dipole pattern which is identified as the NAO.

Similarly to other coupled GCMs, our model in a global warming scenario when considering the full modelled period projects *(i)* a northeastward shift of the NAO centres of action (Ulbrich et al. (1999), Hu et al. (2003), Pausata et al. (2015)) and *(ii)* a very weak but significant positive trend of the NAO (Hu et al. (2003), Gillett et al. (2003), Stephenson et al. (2006)). This suggests that the anthropogenic forcing has a non-null contribution in the NAO evolution. Moreover, in our model the NAO trends continue to interchange in the future. The analysis of the NAO phase distribution reveals an increase of the negative NAO phase number in the future, compared to the past, and a reduction of the positive NAO phase number. Differently, looking at the "high NAO extreme" events ($PC1 > 1.5$) and "low NAO extreme" events ($PC1 < -1.5$), their frequencies respectively rise and decrease in the future.

As far as the NAO impact on tracer transport is concerned, our results show that NAO affects surface tracer concentrations and that tracers concentrate over the Arctic, south Mediterranean and Africa during positive NAO (similarly to Creilson et al. (2003), Eckhardt et al. (2003) and Christoudias et al. (2012) studies). Considering CO-like tracers with constant lifetime and emissions, i.e. disregarding a possible decrease of pollutant emissions in Northern America and Europe, we find that, at the end of the century, tracers over those areas where they are depleted during positive NAO will reduce more, while they will increase over those areas where they are transported to. Therefore, tracers will be transported more efficiently towards the Arctic, south Mediterranean and Africa during positive NAO phases.

Such conclusions are confirmed also by the computation of tracer mixing ratio and transport in the Atlantic sector. Future winter tracer concentrations will increase over Central Europe, south Mediterranean, north Africa, while reduce over north Europe and Greenland due to the increase of "high NAO extreme" events and the prevalence of positive NAO phases. Future tracer transport gets generally stronger over the North Atlantic Ocean and weaker over the Mediterranean region, depleting pollutants from centre-north Europe and causing stagnation over south Mediterranean and north Africa, as already noticed. We remember that these results refer to constant emissions and idealised tracers (i.e. constant decay time).

*Acknowledgements.* The authors wish to extend their gratitude to the MESSy Consortium and the international IGAC/SPARC Chemistry-Climate Model Initiative (CCMI, 2013). The analysed simulations were carried out as part of the Earth System Chemistry integrated Modelling (ESCiMo) project at the German Climate Computing Centre (Deutsches Klimarechenzentrum, DKRZ). DKRZ and its scientific steering committee are gratefully acknowledged for providing the required computational resources.

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

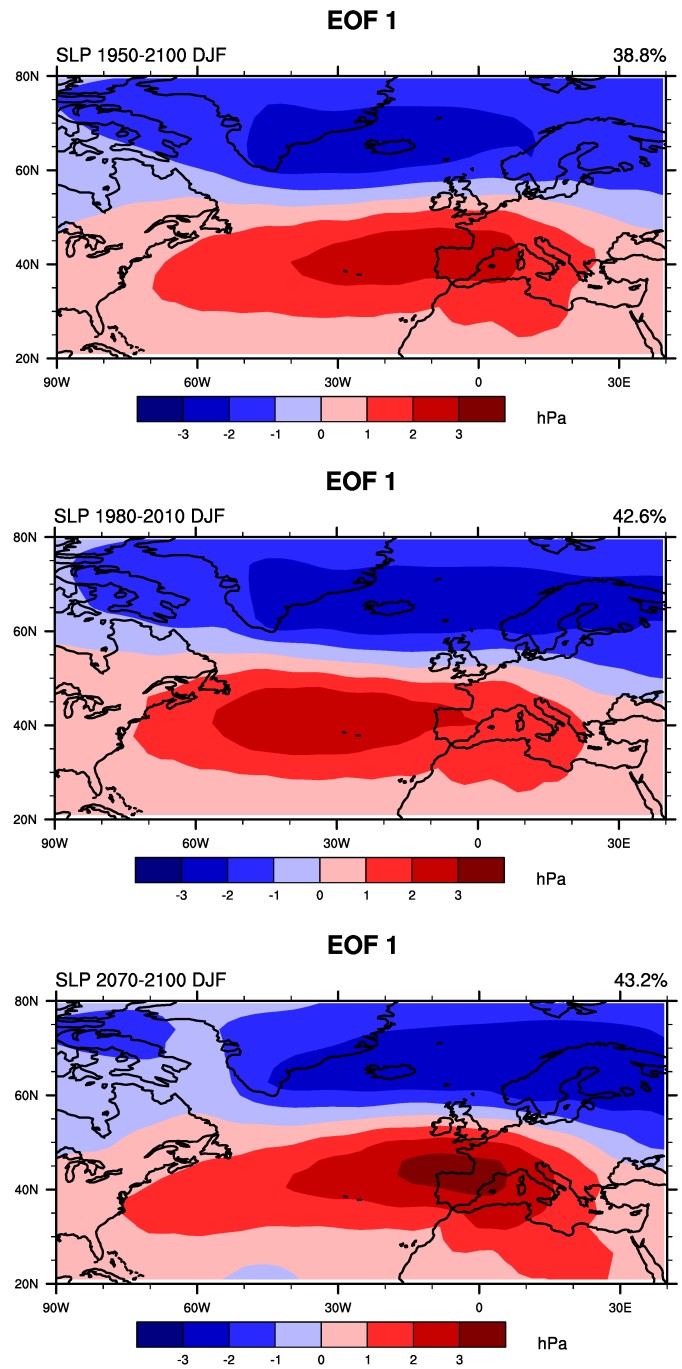

**Figure 1.** Leading empirical orthogonal function (EOF1) of the winter (DJF) mean sea level pressure (SLP) anomalies in the North Atlantic sector (20°N-80°N, 90°W-40°E) of the coupled simulation considering the full period 1950-2100 (*top*), recent past period: 1980-2010 (*centre*), and future period: 2070-2100 (*bottom*). The percentage at the top right of each figure quantifies the total variance explained. The patterns are displayed in terms of amplitude (hPa), obtained by regressing the SLP anomalies on the principal component time series.

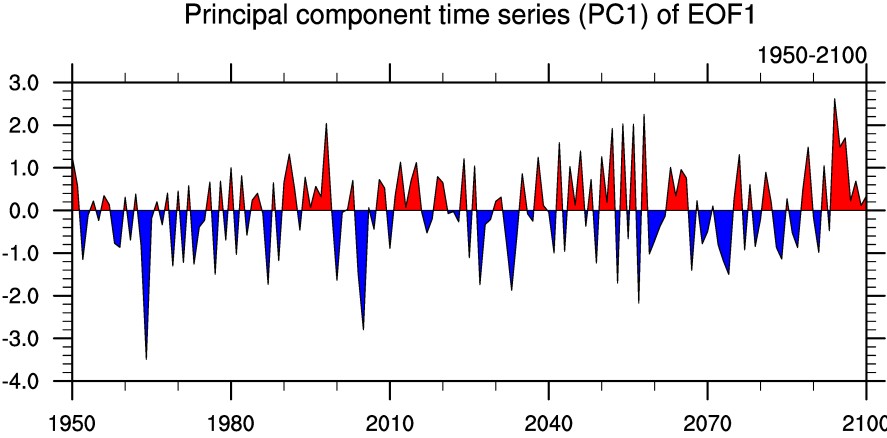

**Figure 2.** Principal component time series (PC1) of the leading empirical orthogonal function (EOF1) of the winter mean sea level pressure (SLP) anomalies for the entire simulation period (1950-2100). The PC1 has been computed after removing the SLP climatology for the recent past (1980-2010).

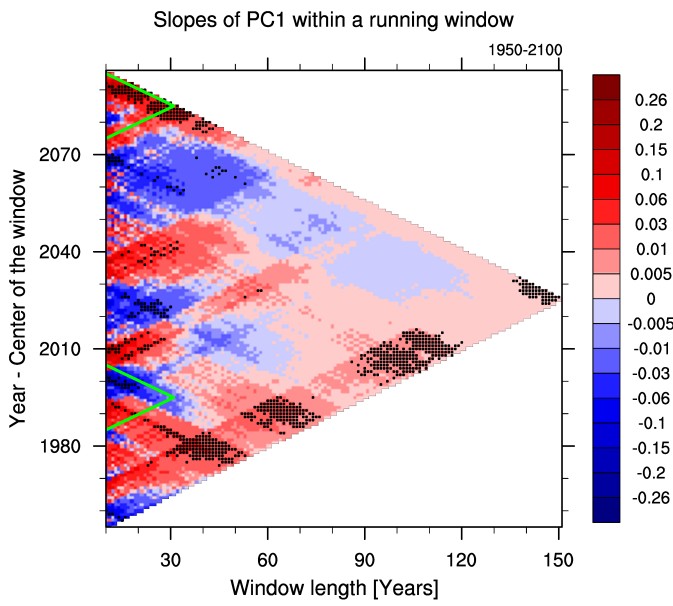

**Figure 3.** Linear regression coefficients of the PC1 based on coupled simulation data computed in sliding windows with variable length for the whole period 1950-2100. Plotted in the x-axis are the window lengths expressed in years, and in the y-axis the central year of the windows. The regression coefficient values are expressed in $\mathrm{hPa/year}$ (see color legend). Points marked with black crosses indicate the 95% level of significance. The green triangles indicate the areas of the two periods recent past and future.

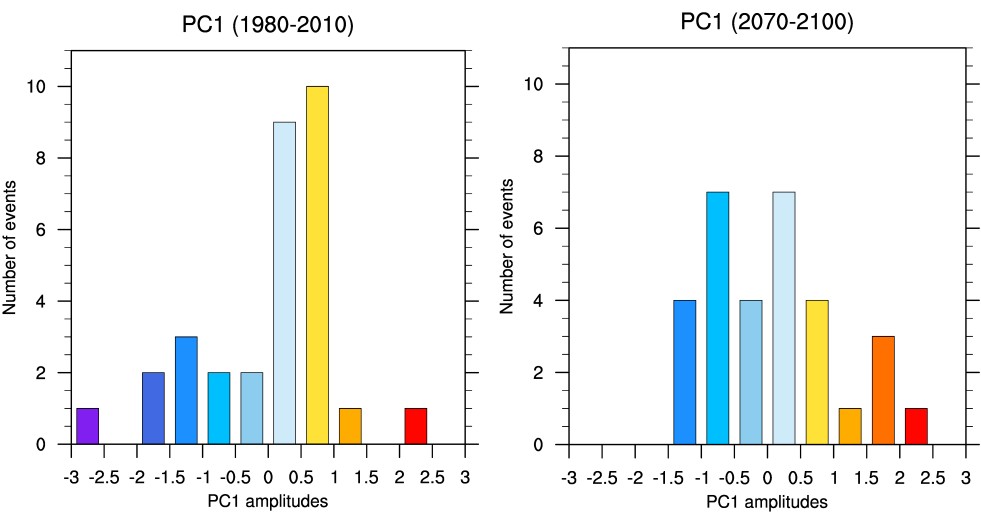

**Figure 4.** NAO phase number distributions, computed in the recent past (*left*) and future (*right*) periods.

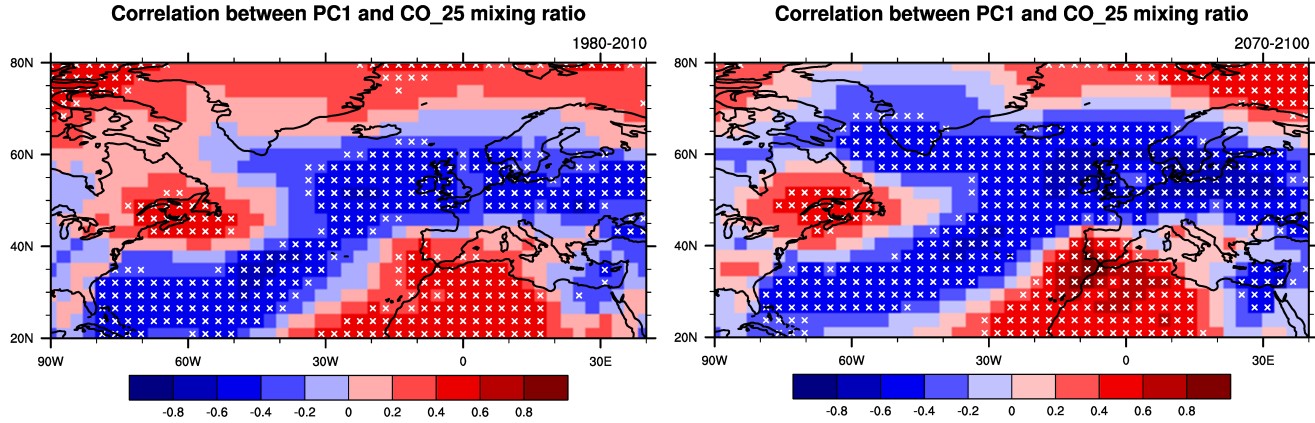

**Figure 5.** Correlation between the winter seasonal CO_25 mixing ratio anomalies at the surface level and the PC1 of SLP computed with the coupled simulation data for the recent past (*left*) and future (*right*) periods. Points marked with a white cross indicate local significance at the 95%.

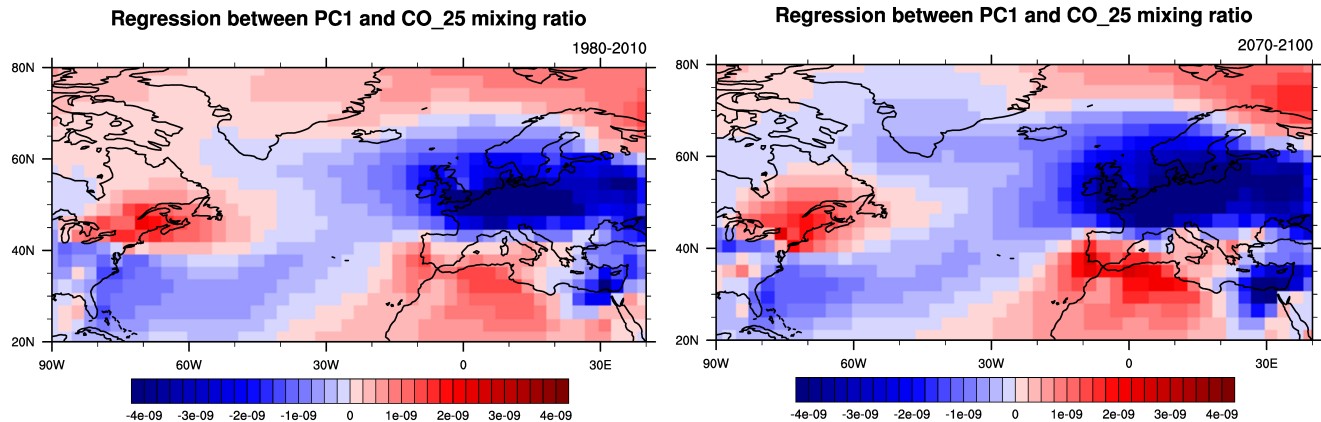

**Figure 6.** Regression between the winter seasonal CO_25 mixing ratio anomalies at surface level and the PC1 computed for the recent past (*left*) and future (*right*) periods.

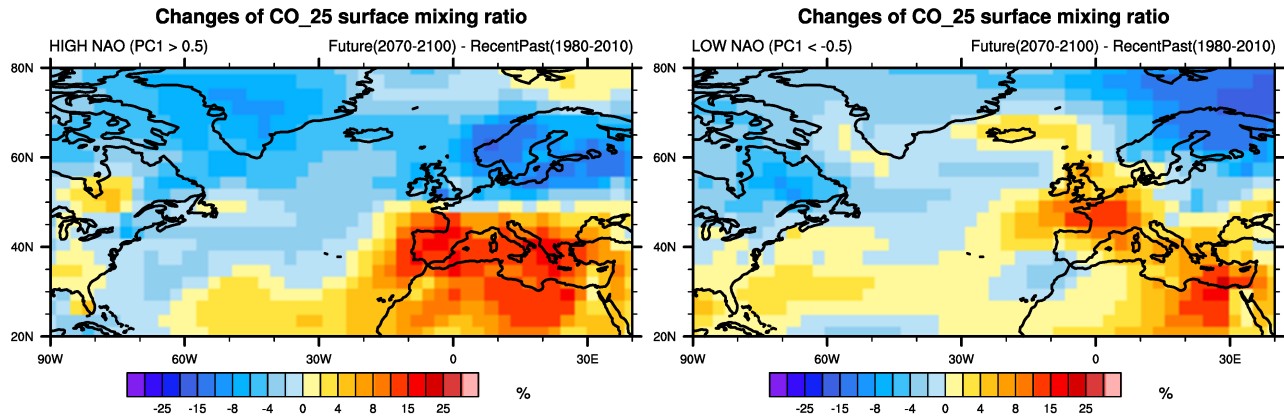

**Figure 7.** Differences between future (2070-2100) and recent past (1980-2010) temporal averages of CO_25 winter surface mixing ratio, both in the case of high NAO (PC1 > 0.5) (*left*) and low NAO (PC1 < -0.5) (*right*). More precisely, plots show the results of $[(CO\_25_{ave}^{fut} - CO\_25_{ave}^{past})/CO\_25_{ave}^{past}] \times 100$, so the colobars indicate the pecentages.

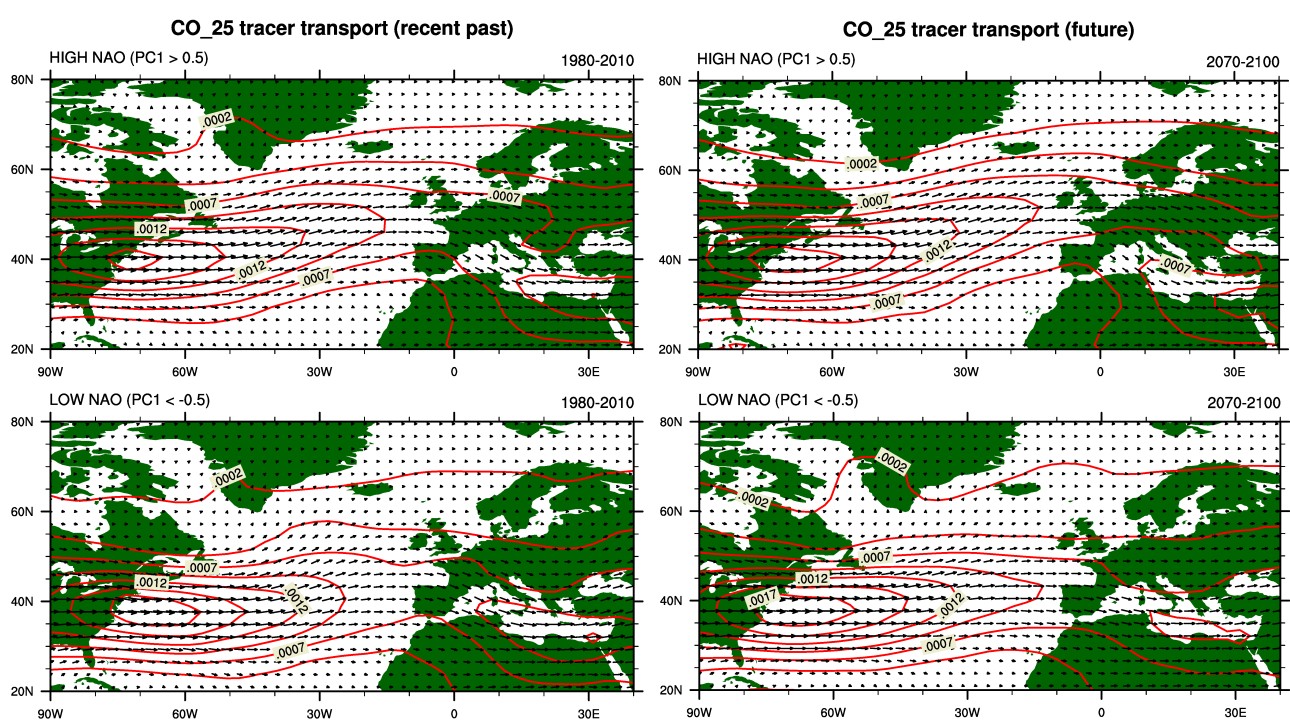

**Figure 8.** Temporal averages of vertically integrated CO_25 tracer transport vectors for winters with high NAO (PC1 > 0.5) (*top*) and low NAO (PC1 < -0.5) (*bottom*), both in the recent past (*left*) and in the future (*right*).