# Peer review of "Figure 1: Correlation between the winter seasonal $CO_{50}$ mixing ratio anomalies at the surface level and the PC1 of SLP computed with the coupled simulation data for the period (A) (top) and (B) (bottom). Points marked with a white cross indicate local significance at the 95%."

_Atmospheric Chemistry and Physics, 2016_

## Referee Comment (RC1) · Anonymous Referee #1 · 23 Jun 2016

Using a fully coupled general circulation model, the authors investigate potential NAO shifts in the future and its impacts on pollutants dispersions. The authors simulate the period 1950-2100. The results show a significant but weak positive NAO and strengthening of the correlation between NAOI(PC1) and CO mixing ratio. Based on that, the authors conclude that under global climate change scenario local air quality conditions in Europe and Northern Africa will become more extreme.

The manuscript is pretty easy to follow, although the presentation should be improved: in the introduction the authors mostly make a list of the papers addressing the relationship NAO and pollution and it lacks of a proper discussion where their results are contextualized within the outcome of previous studies. Further, I think the authors never mentioned the aerosol scenario adopted (MFR, CLE?)

I found the manuscript a bit poor in terms of content and analysis done. The authors only show an EOF analysis and a correlation analysis. I think that after adding further analysis, it may be suitable for publication in ACP.

**Major comments:**

1) I found the final statement that "under a global climate change scenario local air quality conditions over Europe and North Africa, influenced by North Atlantic teleconnection activity, will become more extreme" unsupported.
   The authors also state at the end of pag. 7: "At the same time it seems that the region over the American east coast will be characterized by concentrations of pollutants in a range similar to the past, with respect to the NAO activity." How can they possibly say that just through a correlation analysis? Could the authors explain why an increased correlation between PC1 and CO means that there will be more extremes? The correlation increase in absolute values also in Northern Europe, does that mean it gets more (or less) extreme there too?
   An analysis of the extreme should be done to make that point and would also add more strength to the study. Also a regression analysis can also provide some insights on changes in the relationship between the PC and CO concentrations.
   The authors should also try to understand why the correlation gets stronger in the future. Interestingly, there is a shift of the centers of action but the PC/CO correlation pattern doesn't change. This should be discussed and possibly explained.

2) Another question that I think it would be interesting to address is how the intercontinental transport of pollution changes in the future.

3) I would suggest plotting the PC1 time series for the entire 1950-2100 period. However, the EOF analysis should be done after removing the climatological SLP climatology (e.g. 1980-2010, or 1950-1979). Otherwise if the authors calculate the EOFs without removing any climatology, the PC1 mean of the entire period would be zero. The authors could also plot a pdf of the NAO/PC1 events to see how the distribution (extreme) changes.

These are only few suggestions but I am sure the authors can come up with some ideas and new analysis to perform in order to make the paper more interesting and suitable for the readership of ACP.

4) I am very surprised to see that the coupled simulation shows a similar NAO trend as the reanalysis. The model has its own internal variability therefore is not to be expected at all to show the positive NAO trend in the 80s and beginning of the 90s. It must be a coincidence and should not be presented as an evidence that the model is performing well because resemble the reanalysis.

**Minor comments**

- I would suggest to use superscript rather than subscript for $CO^{25}$ and $CO^{50}$.

- PAG 1 LL15-16: The authors write: "*The NAO, defined as the surface pressure difference between the Azores high and Icelandic low, influences weather conditions (Hurrell, 1995).*"
This is not the definition of the NAO but rather the NAO Index. By definition the NAO is an oscillation a swing between pressure systems not a difference.
The NAO was not discovered by Hurrell, who was the one who introduced the NAOI. Walker in the late 20s was the first to discover it. The reference for the NAO should be:

  Walker, G. T. and Bliss, E. W.: World Weather, V. Mem. R. Meteorol. Soc., 4, 53–83, 1932.

- PAG 4 L28 change data to scenario (the RCPs are scenarios).

- PAG 4 LL29-34: I don't think this is needed. I think is pretty obvious that a coupled simulation is needed for future scenarios and is better than using prescribed SST. In theory you could use SSTs from another model but it's not an optimal solution. Hence, the coupling is the best option and is well known.  If the authors weren't coupling it then you should justify it.

- PAG 5 L12-21 specify the years you are performing the EOF analysis. The 38.8% explained variance is for the entire time series?

- PAG 5 L25: The Eastward/NE shift of the centers of action in future climate is also shown in Pausata et al., ACP, 2015 using another technique.

- PAG 5 L28: change "does not reflect" to "is not able to capture"

- PAG 6 L33 I would avoid using New York to characterize the region. I would just add "of the NORTH American east coast")

- The manuscript has very few figures hence the authors can add the CO50 plot into fig. 4 otherwise what is even the point of saying that you have used the CO50 if you don't show it and the results are the same as CO25?

- PAG 7 L22 please, spell the period out or refer to the experiment as future/reference etc. Avoid using period B or period A.

- PAG. 8 L8  report the citations here as well. In any case a discussion is missing and should be added to the manuscript.

- The authors should take into account in their (future) discussion that in the future the aerosol concentrations will likely decrease and this should anyway lead to a better air quality even over Mediterranean countries. This should be discussed.

---

## Referee Comment (RC2) · Anonymous Referee #2 · 5 Jul 2016

This paper analyzes recent and future trends of the effect of the North Atlantic Oscillation on tracer transport using idealized CO-type tracers in a general circulation model.

While this sounds ideally like an interesting topic, in fact there may not be much here. Long-term trends in the NAO are rather small. While there appears to be a positional change in the NAO, the overall pattern of correlations between the NAO and the tracer appear similar between 1980-2010 and 2070-2100. At this point I'm not at all convinced the future changes in transport due the NAO are important enough to publish (see point 1 below). At the minimum the paper needs major revisions in order to be acceptable for publication.

1. The correlation pattern between PC1 and CO25 look rather similar between the present-day and future periods, although they differ in detail. The correlation maps

however, really do not indicate the magnitude of the differences. It would be better to regress the NAO against the CO25 concentration during the two periods and determine differences between CO25 concentrations. Are these really large? In my opinion this extension and its significance is necessary for publication.

2. Differences in tracer concentration due to the NAO could be due to both trends in NAO or in changes in the EOF pattern. I believe the changes in Figure 4 are due to changes in the pattern. However, the authors should also examine changes in the tracer concentration due to changes in the NAO trend (or are these not important?). The authors concentrated on changes in the NAO trend in the first part of the manuscript so the importance of temporal changes should be addressed.

3. When computing changes in the NAO trend I assume the authors are allowing for variations in the pattern of the 1st EOF depending on the time period. This should be made very clear (in section 3.2). My feeling is it would be better to look at trends in an invariant NAO pattern.

4. I am puzzled between the similarity in pattern between the slopes of the model generated PC1 and the observational generated pattern over the historical period. Has this been seen in other general circulation models considering a substantial portion of the NAO is forced by atmospheric variability? The authors suggest that similarities in patterns and slope are in good agreement validating the ability of the model to correctly simulate the NAO. However, why does one expect agreement?

5. Finally, I'm somewhat concerned about the shift in position of the NAO between future and present climates (Figure 1). Could this shift and the corresponding changes in tracer correlation (Figure 4) simply be due to interdecadal variability? This distinction seems somewhat important as the significance of the change in pollution with future changes in the NAO depends on the fact that the shift in the NAO is climate induced. I feel the authors should try to demonstrate that this shift is not simply due to interdecadal variability.

---

## Author Comment (AC1) · 19 Sep 2016

We thank the anonymous referee for the review and comments, which have been very useful to improve our manuscript.

Here below, our replies to the referee's comments.

In the introduction we have provided several references regarding the NAO influence on gas pollutants [PAG 2 LL 1-15] and aerosol concentrations [PAG 2 LL 16-22] in order to give a clear view to the reader about the effects of the NAO on tracer transport. However, we will also contextualize better our results in the conclusions, as suggested by the referee.

The simulation analysed in this work does not include interactive aerosols nor aerosol scenarios, but only a climatology of them is present in the model simulation [PAG 4 L 26], as described in Jöckel et al. (2016), thus our study does not look into aerosol effects. We will specify this better in the Methodology Section.

**Major Comments**

1. We agree with the points raised by the referee. From the correlation analysis we estimated where European and Eastern USA CO-like tracers concentrate/deplete during positive/negative NAO phases. Therefore, higher correlations (in absolute value) imply that an increase/decrease of the PC1 will drive a higher/lower stagnation of such pollutants on specific regions. For this reason we suggest that changes in the NAO index (PC1) could impact certain locations. On the other side, this work is related only on the transport of CO-like tracers with constant lifetime and emissions and does not account for a possible (and probable) decrease of pollutant emissions both in Northern America and in Europe. Therefore, as suggested by the referee, we will reformulate parts of the text to be more coherent with the rest of the manuscript.

Indeed, as written in pag. 7, the anti-correlation area over northern Europe is stronger (i.e. higher absolute values) in the future than in the past [PAG 7 LL 21-22]; we will improve our explanation in the light of the new regression analysis (see next paragraph).

As suggested by the referee, we have now performed the analysis of the extremes. More precisely, we have computed the temporal averages of CO 25 winter surface concentrations for high and low NAO events, both in the recent past (1980-2010) and in the future (2070-2100). We have chosen to define "high NAO" and "low NAO" those (winter) periods when PC1 is higher than 0.5 and lower than -0.5, respectively. In this way, we have obtained 12 high and 8 low NAO phases in the recent past and 9 high and 11 low NAO phases in the future (thus, averages have always been computed over at least 8 values). In order to investigate how the CO\_25 concentrations change in the future, we have plotted the differences between future and recent past concentrations during high and low NAO periods (Fig. 1). We can observe that in the future, during high NAO (Fig. 1, left), concentrations increase by 10% over north Africa and Mediterranean and even by 15% over some areas of the Iberian Peninsula, Greece and Aegean Sea. Concentrations are lower than in the past over northern Europe and Greenland (in the range down to -10%). On the other hand, during low NAO (Fig. 1, right) we find that CO 25 concentrations increase over north-east Africa and west-centre Europe (up to 15%) and decrease between the north Scandinavian and the Arctic and over small areas on North America and Atlantic Ocean (down to -10%). Therefore, this analysis provides further evidence for our conclusions and will be added to the manuscript (in Section 4) to add more strength to our study.

Following the suggestion of both referees, we have computed the regression between the PC1 and CO\_25 mixing ratio in the recent past (Fig. 2, left) and in the future (Fig. 2, right). The patterns and the evolution at the end of the century observed with the regression analysis are very similar to what we have already noted with the correlation plots. To corroborate our analysis we will add these new plots in our manuscript.

Our results show a shift of the NAO centres of action, in agreement with the results obtained by Ulbrich et al. (1999) and Hu et al. (2003) for a climate change global warming scenario [PAG 5 LL 24-25]. However the investigation of NAO shift falls outside the scope of our study, which focuses on NAO effects on tracer transport [PAG 8 LL 13-14].

2. Following the hint of studying the pollutant transport changes in the future, we have computed (and we will add in the manuscript) the vertically integrated tracer transport vector, defined as:

$$\vec{Q} = \frac{1}{g} \int_0^{ps} r \vec{u} dp \tag{1}$$

where r is the concentration of CO\_25 in mol/mol,  $\vec{u}$  is the horizontal wind speed, p is the pressure, ps is the surface pressure and g is the gravitational acceleration. Also in this case, we have computed the temporal averages of  $\vec{Q}$  of all winters with high and low NAO (defined as in the previous point) both in the recent past and in the future. In Fig. 3 we can observe that during high NAO (top) the northward shift of CO\_25 transport in the future is more pronounced over the North Atlantic Ocean, from the Northern America coast towards Ireland, while it gets weaker over south Greenland, Mediterranean and west Europe. During low NAO (bottom), the eastward CO\_25 transport over the North Atlantic Ocean extends farther eastwards in the future, while it decreases over the Mediterranean Sea; differently from the high NAO case, transport gets slightly stronger over south Greenland in the future. The primary findings of the transport of CO\_25 which gets generally stronger over the North Atlantic Ocean and weaker over the Mediterranean region is in line with the changes observed in the correlation and regression plots.

3. We agree with the referee and will be adding the plot of the entire PC1 (1950-2100), computed after subtracting the SLP climatology of 1980-2010. The result is shown in Fig. 4.

In order to study how the distribution of (extreme) NAO phases will change in the future, we have computed the frequencies of (winter seasonal) NAO phases in the recent past and in the future. The histograms with frequencies for 1980-2010 and 2070-2100 are shown in Fig. 5 (left and right respectively). We have preferred not to compute the probability density function (PDF) of NAO phases given the low statistics. In the recent past (Fig. 5, left) the distribution covers a large PC1 interval ([-3; 2.5]) and the number of NAO phases is at most 3, except in the interval [0; 1] where it is clearly higher (equal to 9 and 10). Differently, in the future the distribution is skewed toward positive values of PC1 (the interval is [-1.5; 2.5]), with number of NAO phases between 4 and 7 in the interval [-1.5; 1] and between 1 and 3 in the interval [1; 2.5]. Thus, at the end of the century the number of negative NAO phases is increased (15 in the future vs. 10 in the past), vice-versa, the number of positive

NAO phases is decreased (16 in the future vs. 21 in the past). However, the number of "high NAO extreme" events (PC1 > 1.5) increases in the future (4 in the future vs. 1 in the past), while the number of "low NAO extreme" events (PC1

Figure 1: Differences between future (2070-2100) and recent past (1980-2010) temporal averages of CO\_25 winter surface mixing ratio, both in the case of high NAO (PC1 > 0.5) (*left*) and low NAO (PC1

---

## Author Comment (AC2) · 19 Sep 2016

We thank the anonymous referee for the review. It is a matter of debate that if a manuscript shows "negative results" does not deserve to be published and this has also been discussed in recent time (Knight (2003), Fanelli (2011)). Nevertheless, in this study not only a marked shift of the NAO centers of action is estimated for future scenario, but also potential changes in the distributions of the principal components describing the NAO, changes in correlations and also (now additionally added) transport patterns are shown. Therefore, we strongly believe that this in itself merits publication.

Here below, our replies to the referee's comments.

**Replies to Referee's Comments**

1. Following the suggestion of both referees, we have significantly improved the analysis of the CO_25 concentration changes between the periods 1980-2010 and 2070-2100. In particular, we have computed the temporal averages of CO_25 winter surface concentrations for high and low NAO events following the criteria written in our reply to referee #1, point-1.

   Moreover, we have computed the regression analysis (Fig. 2 in the reply to referee #1) that will be added in our manuscript.

2. Indeed, differences in tracer concentration due to the NAO could be due to both trends in NAO or changes in the EOF pattern. The computation of changes due to NAO trends contributes to our understanding of temporal concentrations changes by weighing the correlation results due to patterns with the results of the analysis of trends, performed in Section 3.

3. Actually, NAO trend values reported in Fig. 3 of our manuscript are obtained from the PC1 computed for the entire period (150 years), i.e. given the long PC1 we computed the trends in windows sliding along this PC1 series. We will clarify this in Section 3.2 as it was not clear in the original manuscript.

4. As replied to referee #1, the agreement between the coupled simulation and the observations is not expected. In order to avoid any misunderstanding, we have decided to eliminate the analysis considering the "NAO-PC-based index" and Fig. 2 from the manuscript (see also our answer to referee #1, point-4).

5. A climatological timescale (30 years) for the two periods (recent past and future) has been chosen to reduce the interdecadeal variability, as feared by the referee. To corroborate the assertion that the NAO shift is climate induced, rather that due to interdecadal variability, we have chosen different climatological timescales of 30 years for the past and future and computed the decadal EOF1 for both, i.e. 1950-1979, 1960-1989, 1970-1999, 1980-2009 in the past and 2040-2069, 2050-2079, 2060-2089, 2070-2099 in the future. The results in Fig. 1 show differences between the two climatological periods, but they do not between any of the decadal timescale within each period. Thus, we deduce that the changes observed between the past NAO pattern and the future NAO pattern are climate induced and are not simply due to decadal variability. This information will be added to the manuscript and the figures to the electronic supplement.

*References*

- Knight, Nature 422, 554-555, doi:10.1038/422554a, 2003.

- Fanelli, D.: Negative results are disappearing from most disciplines and countries, Scientometrics 90, 891-904, 2012.

[Figure]

Figure 1: Leading empirical orthogonal function (EOF1) of winter mean sea level pressure (SLP) anomalies of the coupled simulation. From top to bottom and left to right, the leading EOFs correspond to the 30 years periods 1950-1979, 1960-1989, 1970-1999, 1980-2009 (past), and 2040-2069, 2050-2079, 2060-2089, 2070-2099 (future).

---

## Author Response (AR1)

Dear Editor,

below we describe the changes to the manuscript.
Since the new manuscript is quite different from the previous one, we do not highlight the differences in the text, but rather we specify the changes using the notation: [PAG XX LL XX]. Please, note that the references to pages and lines refer to the new version of the manuscript (differently from the replies to Referees).

Changes summary:

1. The Introduction is basically the same, with only minor changes.

2. The Methodology is basically the same, although minor changes have been implemented and the last paragraph for the description of the vertically integrated tracer transport vector computation has been added.

3. Section 3 has been partially modified to introduce the new analysis.

4. Section 4 has been largely changed. Now there are two subsections in order to include the new analysis performed to satisfy the Referees' comments.

5. The Conclusions and the Abstract have been modified accordingly to the new analysis.

6. Now there are 8 figures in total (please, note that only 3 figures out of 4 of the previous version are included in the new manuscript).

In order to facilitate the work of the Editor, below the Referees' comments are listed (as *[...]*) followed by the notes about our changes.

- - - - - - - - - - - - - - - - - - - **REFEREE #1** - - - - - - - - - - - - - - - - - - - -

*[...it lacks of a proper discussion where their results are contextualized within the outcome of previous studies.]*
In the conclusions we have contextualized better our results and mentioned the previous studies cited in the introduction [PAG 10 LL 3-5 and LL 11-13].
*[I think the authors never mentioned the aerosol scenario adopted (MFR, CLE?).]*
We have specified better that the simulation uses a climatology of aerosols in the Methodology section [PAG 4 LL 26-28] and conclusions [PAG 9 L 32].

**Major Comments**

1. *[...How can they possibly say that just through a correlation analysis? Could the authors explain why an increased correlation between PC1 and CO means that there will be more extremes? ...]*
   We have explained the role of the correlation computation in our analysis at PAG 7 LL 17-21.

   *[An analysis of the extreme should be done to make that point and would also add more strength to the study.]*
   We have added a new analysis of the extremes. We have defined "high NAO" and "low NAO" as (winter) periods with PC1 higher than 0.5 and lower than −0.5, respectively,

and we have computed the temporal averages of CO_25 winter surface mixing ratio. This analysis has been included in the new Subsection 4.2 [PAG 9 LL 2-16] together with the new Figure 7. Please, note that the old Section 4 now has two subsections: "4.1 Correlation and regression analysis" and "4.2 Tracer concentration and transport changes".

*[Also a regression analysis can also provide some insights on changes in the relationship between the PC and CO concentrations.]*
We have computed the regression between the PC1 and CO_25 mixing ratio in the recent past and in the future. Thus, we have added the new regression analysis and the new Figure 6 after the correlation analysis [PAG 8 LL 12-27].

*[... there is a shift of the centers of action but the PC/CO correlation pattern doesn't change. This should be discussed and possibly explained.]*
We answered in the Referee reply that the investigation of the NAO shift falls outside the scope of our study, which focuses on NAO effects on tracer transport.

2. *[Another question that I think it would be interesting to address is how the intercontinental transport of pollution changes in the future.]*
We have computed the vertically integrated tracer transport vector in order to study the pollutant transport changes in the future and we have added this new analysis in the Subsection 4.2 [PAG 9 LL 17-27] together with the new Figure 8. The description of the computation has been written in the Methodology section [PAG 5 LL 11-15].

3. *[I would suggest plotting the PC1 time series for the entire 1950-2100 period. However, the EOF analysis should be done after removing the climatological SLP climatology (e.g. 1980-2010, or 1950-1979).]*
We have added the new Figure 2 of the entire PC1 (1950-2100) series, computed after subtracting the SLP climatology of 1980-2010, in the Subsection 3.1 [PAG 6 LL 13-14].

*[The authors could also plot a pdf of the NAO/PC1 events to see how the distribution (extreme) changes.]*
We have included the new analysis of the NAO phase distribution and the new Figure 4 at the end of the Subsection 3.2 [PAG 6 LL 32-34 and PAG 7 LL 1-6], after the trend analysis. As we wrote in the Referee reply, we preferred to describe the distribution through an histogram instead of computing the PDF given the low statistics.

4. *[I am very surprised to see that the coupled simulation shows a similar NAO trend as the reanalysis. The model has its own internal variability therefore is not to be expected at all to show the positive NAO trend in the 80s and beginning of the 90s. It must be a coincidence and should not be presented as an evidence that the model is performing well because resemble the reanalysis.]*
We have removed the text regarding the NAO trend observations (and the relative plot) from the Subsection 3.2. Thus, the only figure showing the NAO trends now is Figure 3.

**Minor Comments**

**Subscript** We have changed the notation to "CO_25" and "CO_50" (according to Eyring et al. 2013, pag. 48-66).

**PAG 1 LL 15-16** We have modified the text [PAG 1 LL 16-18] as: "It is a swing between two pressure systems, the Azores high and Icelandic low, which redistributes atmospheric masses between the Arctic and the subtropical Atlantic influencing weather conditions (Walker et al. 1932)."

**PAG 4 L 28** We have changed the text as requested from "data" to "scenario" [PAG 4 L 29].

**PAG 4 LL 29-34** We have preferred to leave this paragraph [now PAG 4 LL 32-35, PAG 5 LL 1-2] because here we explain that coupled models are better than GCM forced with SST to simulate the NAO phenomenon itself, besides future scenarios.

**PAG 5 LL 12-21** We have specified the years considered for the EOF analysis [PAG 5 L 25-26].

**PAG 5 L25** We have added the new reference "Pausata et al. 2015" [PAG 6 L 6].

**PAG 5 L28** We have changed the text as requested from "does not reflect" to "is not able to capture" [PAG 6 L 10].

**PAG 6 L33** We have changed the text as requested [PAG 7 LL 22-23], but we have preferred to write "Northern America east coast", instead of "North American east coast" as suggested by the Referee.

**CO_50 figure** As we have performed other plots which have been added in the manuscript (Figures 2, 4, 6, 7, 8), we have decided to leave the figures for CO_50 in the electronic supplement. The reason to show the analysis of CO_50 is explained at PAG 7 LL 13-16 and LL 25-27.

**PAG 7 L22** We have changed the text using "recent past" and "future", instead of "period A" and "period B" as requested.

**PAG 8 L8** We have improved our discussion and added the citations in the conclusions [PAG 10 LL 3-5].

**aerosols concentrations** We have specified in the text that the simulation does not account for a possible decrease of pollutant emissions and that we focus on the transport of CO-like tracers [PAG 8 LL 31-35 and PAG 10 L 23].

- - - - - - - - - - - - - - - - - - - - **REFEREE #2** - - - - - - - - - - - - - - - - - - - - -

1. *[...It would be better to regress the NAO against the CO25 concentration during the two periods and determine differences between CO25 concentrations...]*
We have added the analysis of the CO_25 mixing ratio future changes (Figure 7) and the regression analysis (Figure 6). Please, see also point-1 Referee #1.

2. *[...the authors should also examine changes in the tracer concentration due to changes in the NAO trend...The authors concentrated on changes in the NAO trend in the first part of the manuscript so the importance of temporal changes should be addressed.]*
As we wrote in the Referee reply, differences in tracer concentration due to the

NAO could be due to both trends in NAO or changes in the EOF pattern. The computation of changes due to NAO trends contributes to our understanding of temporal concentrations changes by weighing the correlation results due to patterns with the results of the analysis of trends, performed in Section 3.

3. *[When computing changes in the NAO trend I assume the authors are allowing for variations in the pattern of the 1st EOF depending on the time period. This should be made very clear (in section 3.2). My feeling is it would be better to look at trends in an invariant NAO pattern.]*
   We have clarified that the trend values in Figure 3 are obtained from the PC1 computed for the entire period of 150 years [PAG 6 LL 16-18].

4. *[I am puzzled between the similarity in pattern between the slopes of the model generated PC1 and the observational generated pattern over the historical period...Why does one expect agreement?]*
   As written in point-4 Referee #1, we have removed the old text regarding the NAO trend observations (and the relative plot) from the Subsection 3.2. Thus, the only figure showing the NAO trends now is Figure 3.

5. *[...I feel the authors should try to demonstrate that this shift is not simply due to interdecadal variability.]*
   We have performed a new analysis to show that the NAO shift is climate induced and we have added it in the Subsection 3.1 [PAG 5 LL 30-31 and PAG 6 LL 1-4]. The new Figure has been included in the electronic supplement file.

---

## Author Response (AR2)

We thank the anonymous Referee and the Editor for their review and comments.

Below there are our replies to the Referee's comments. Please, note that all mentioned pages and lines refer to the same version which also the Referee referred to.

**1) P1, L7:**
*"interchange": meaning not clear.*
We changed the old sentence: "Moreover, we find that NAO trends (over periods shorter than 30 years) will continue to interchange in the future." to: "Moreover, we find that NAO trends (computed over periods shorter than 30 years) will continue to oscillate between positive and negative values in the future."

**2) P1, L10:**
*"increased pollutant depletion": awkward phrasing.*
We changed the old sentence: "...while a wider part of north Europe will benefit from increased pollutant depletion." to: "...while a wider part of north Europe will benefit from lower pollution."

**3) P1, L13:**
*What do the authors mean by "teleconnection activity"?*
The NAO is one of the most prominent teleconnection pattern over the Northern Hemisphere, besides the Pacific-North American (PNA) pattern (Hurrell et al. 2003). Thus, in our manuscript at P1 L13 we referred to this fact. Nevertheless, we chose to delete such sentence for the reason explained in the next point.

**4) P1, L13:**
*The last sentence in the abstract seems a bit strong, that "local air quality conditions over Europe... will become more extreme" particularly since the authors suggest in the sentence above that portions of Europe will see less pollution.*
Saying "will become more extreme" we mean that the air quality conditions will improve over north Europe while will worsen over south Europe and north Africa. However, since such sentence describes (in a shorter way) what is already written some lines before, at P1 L8-10, we decided to delete it.

**3) P3, L17-21:**
*Repetitive with P2, L25-28.*
We agree with the Referee. We have maintained the sentence at P2 L25-26 and we have deleted the sentences later (P2 L26-28). On the other hand, we have left the paragraph at P3 L17-21.

**4) Section 2:**
*I assume sea-ice is also modeled? Please mention explicitly.*
Yes, the sea-ice distribution and thickness is interactively calculated by the ocean model. We added this information in the Methodology section (P4 L16).

**5) P4, L13:**
*"the development of the climate": delete "the development of"*
Now the sentence is: "...'coupled simulation', simulates the climate covering the period 1950–2100."

**6) P4, L27:**

*"climatology of aerosols" Do the aerosols change with time? This has been implicated as important in simulating the NAO.*

The aerosol distributions used in this work are based on a monthly climatology, therefore, they do have a monthly variation, but it is kept constant throughout the simulated years. We added such information in the Methodology section (P4 L27).

**7) P5, L4:**

*"all associated feedbacks": is the carbon cycle modeled?*

No, it is not. With the expression "all associated feedbacks" we actually referred to the domains mentioned in the same sentence. However, to make more clarity we have chosen to delete it and to specify also that the coupling between ocean and atmosphere is a dynamical one. Thus, the sentence now is: "Our model is one of the first to include a full dynamical ocean-atmosphere coupling, stratospheric circulation in conjunction with online chemistry and anthropogenic emissions, thus providing state-of-the-art simulation capability of the phenomenon and potential impacts."

**8) Equation 1 and Figure 8:**

*What are the units? With pressure in pascals and the mixing ratio in mol/mol the units are a bit strange...*

To compute the vertically integrated tracer transport vector we have followed the same method used by Hurrell (1995), considering the mixing ratio of $CO$ instead of the specific humidity. The unit of Eq. (1) is $\frac{mol}{mol}\frac{kg}{m \cdot s}$, analogously to that in Hurrell (1995). We added the reference to Hurrell (1995) at P5 L12 and the unit in Fig. 8.

**9) P6, L4-5:**

*The first figure in the supplement indeed suggests that the NAO pattern shifts between the present and future climate. However, the clause, "but they do not between any of the decadal timescale within each period" needs amending. This sentence is not clear and the individual panels are shown on a basis of 30 years not in decades. Please clarify.*

We modified such sentence as: "The results (shown in the electronic supplement) exhibit differences between the two periods, past and future, but they do not between any of the climatological timescale within each period. Thus, the changes observed for the past and future NAO patterns are not due to decadal variability but rather they are climate induced."

**10) P6, L16:**

*Perhaps make clear the linear regression coefficient is with respect to time.*

We mentioned this and now the sentence is: "To investigate the NAO temporal variability and trends we calculate, considering sliding windows, the linear regression coefficients with respect to time of the PC1 computed for the entire 150-years simulation (Fig. 2)."

**11) P6, L24:**

*Please give units.*

We augmented the text: "$(2.99 \times 10^{-3} \pm 0.95 \times 10^{-3})$ $[1/year]$".

**12) P6, L31:**

*The short-term trend in the NAO at the end of the century does not look significant in the*

*sense that if extended the simulation by 10 years one would get a different result. Thus, it is not clear of the importance of this line.*

The Referee is right. However, this sentence describes one of the two triangles marked in Fig. 3, and the lower triangle is described just in the sentence before. We think that such line could be left for a complete description of Fig. 3.

**13)**

*It is perhaps worth stressing that the results in Figure 4 are not surprising, but in fact are consistent with the positive trend: an increase in high level events in the future and a decrease in low level events.*

Yes, thanks. We modified the end of Section 3.2 (at P7) as follows: "However, the "high NAO extreme" events (PC1 > 1.5) are more frequent in the future (4 in the future vs. 1 in the past), while the number of "low NAO extreme" events (PC1 < −1.5) decreases (0 in the future vs. 3 in the past), and such results are consistent with the future positive trend commented before."

**14)**

*The units on Figure 6 are not given. One should regress CO against the principal component and then multiply by the standard deviation of the principal component to get typical changes in CO (in ppb) due to NAO variability.*

The regression is actually computed setting PC1 as the independent variable $x$ and $CO$ as the dependent variable $y$. However, the normalized PC1 is used to compute the regression, thus the standard deviation is equal to one. We have chosen to extend the sentence at P8 L12-13: "we compute the regression between $CO\_25$ mixing ratio and PC1 (Fig. 6)." to: "we regress the $CO\_25$ mixing ratio against the normalized PC1 (Fig. 6)."

We added in the caption of Fig. 6 the unit $mol/mol$, i.e the dimension of the dependent variable as the independent variable has been normalized.

*The correlation and regression should look exactly the same (except for units) so it is not necessary to show figure 5, only figure 6.*

We agree with the Referee, thus, we decided to move the correlation figure (Fig. 5) to the supplement file.

*The significant regions, however, should be marked in Figure 6. The difference map between the future regression and the present-day regression should be shown. These changes would act to limit the rather lengthy explanations in the text. The text describing figures 5 and 6 should be shortened.*

We marked the significant regions on the regression plots as requested by the Referee.

We added the difference plot (Fig. 1 in this document) in the electronic supplement so that the manuscript has kept its structure.

As we moved Fig.5 to the electronic supplement the text has been shortened, as requested.

**15) P7, L19-20; P8, L 13-14:**

*It seems unnecessary to explain exactly what a positive or negative correlation (regression) means.*

As we have reduced the text for Fig. 5 and 6 we have also shortened such explanations (although we have not completely deleted them).

**16)**

*Differences between high and low NAO (Figure 7). This is probably a nice way of showing future changes due to the NAO. However, it is likely that in the future the events catego-*

*rized as high have, on average, a higher NAO index than those characterized as high in the present; similarly and the future events categorized as low in the future are likely less negative than those characterized as low in the present. This alone would explain Figure 7; that is, it is probably not true that the response to the NAO becomes more extreme in the future, just that there are more positive events in the future. This distinction should be clarified.*

We extended the text in order to better describe the differences between future and recent past PC1 amplitudes (see also next paragraph) at P9 L5 and the explanation of Fig. 7 at P9 L13.

*The averages of the future and present high and low NAO indexes that are used in Figure 7 should be given.*

We added the averages of the recent past and future high and low NAO indexes. Thus, the paragraph at P9 L2-4 was changed to: "We obtain 12 high and 8 low NAO phases in the recent past and 9 high and 11 low NAO phases in the future. The averages of the PC1 amplitudes (all computed over at least 8 values) are equal to $-1.38$ in the recent past and $-1.01$ in the future considering the low NAO events and equal to $0.83$ in the recent past and $1.24$ in the future considering the high NAO events. Thus, we note that in the future the events categorized as high will have, on average, a higher PC1 amplitude than those in the recent past and, similarly, the future events categorized as low will be less negative than those in the recent past. Therefore, we find that the number of low/high NAO events will increase/decrease in the future, while the mean PC1 amplitudes will increase in the future in both cases (low and high NAO). "

*Discuss whether the results given in Figure 7 are consistent with the regressions shown in Figure 6 (taken into account the differences in the high and low events). That is, are the results the same one would get assuming linearity or are the high end events getting more extreme?*

We had wrote at P9 L13 that results in Fig. 7 were in line with the results found with the correlation and regression analysis.

**17) Figure 8:**
*Please include the units. Also, showing a plot of future minus present would be helpful in visualizing the differences.*

We added the unit $\frac{mol}{mol}\frac{kg}{m \cdot s}$ in the caption of the Fig. 8. Moreover, we added the plot showing the differences between future and recent past (Fig. 2 in this document) in the supplement file. Consequently, the paragraph at P9 L20-25 has slighlty changed.

**18) P10, L7-8:**
*Please check the wording here. I thought the future showed an increase in the amplitudes of the positive NAO and a reduction in the amplitudes of the negative NAO.*

Yes, it true. However, with the sentence indicated by the Referee (P10 L7-8) we actually consider the changes of the total number of positive and negative NAO phases (and not the PC1 amplitudes) as described in P7 L3-4. To improve the text we changed the lines P10 L7-10 to: "The analysis of the NAO phase distribution reveals an increase of the negative NAO frequencies in the future although with much reduced amplitudes. On the contrary, positive NAO phases do decrease in frequency but increase in amplitude."

**19) Conclusions:**
*Which is more important, the shift in the NAO or the change in its amplitude? (Or maybe the authors could not determine this). However, it seems it should be at least mentioned.*

Indeed, it is difficult to fully extrapolate the impact of shifted NAO and of the change in its amplitude. We reformulated the sentence mentioning that both causes can contribute to the effect we observe (P10 L20).

**20) P10, L16:**
*The "Therefore" here needs some more explanation I think.*
Such sentence is a consequence of what written just before, so we think that rephrasing it should make more clarity. We changed the old sentence "Therefore, tracers will be transported more efficiently towards the Arctic, south Mediterranean and Africa during positive NAO phases." to: "This means that tracers will be transported more efficiently towards those areas which already suffer for bad air quality conditions during positive NAO, i.e. over the Arctic, south Mediterranean and Africa."

[Figure]

Figure 1: Difference between future and recent past regression of $CO\_25$ over PC1.

[revised manuscript text omitted]

---

## Author Response (AR3)

We thank the Editor for his work in reviewing our manuscript.

Our replies can be found below each point.

**1) P1, L9-10:**

*My understanding is that this paper addresses 2 independent points – (i) will there be a future change in the NAO and (ii) will there be a future change in the structure and hence the chemical implications of the NAO as defined as the leading mode of variability? I don't believe that any of your results directly address – "will there on average be a chemical change in future", but what is addressed is "will positive NAO vs negative NAO have different chemical implications in future compared to the present."*

*This needs to be made transparently clear to the reader and a sentence such as "We find that at end of the century during positive NAO phases south-west Mediterranean and north Africa will see higher pollutant concentrations with respect to the past, while a wider part of north Europe will benefit from lower pollution." in the abstract is not very clear in my view – a casual reader may interpret it as saying "We find at the end of the century... a wider part of north Europe will benefit from lower pollution". A possible clarification would be: "We find that at the end of the century south-west Mediterranean and north Africa will during positive NAO phases see higher pollutant concentrations with respect to the past, while a wide part of north Europe will, during NAO phases, see lower pollutant concentrations." (The suggested sentence might seem unnecessarily repetitive, but at least it is transparently clear.)*

*Please consider carefully whether the distinction between (i) and (ii) above is made completely clear throughout the paper as a whole.*

Indeed, the Editor correctly identifies the two independent points in this manuscript: (i) will there be a future change in the NAO and (ii) will there be a future change in the structure and hence the in the tracer transport? In both questions, the NAO was defined as the leading mode of variability. We also did not address any absolute chemical change in the future, but rather the implication on tracer transport of "positive NAO vs negative NAO [...] in future compared to the recent past". As suggested by the Editor, we modified the abstract accordingly.

Additionally, we changed P8 L22-24: "Consequently, at the end of the century south-west Mediterranean and north Africa will suffer from higher pollutant concentrations during positive NAO phases compared to the past, while a wider part of north Europe will benefit from lower concentrations of long range pollutants (associated with improved surface air quality) during the positive NAO phases with respect to the past.".

Moreover, we clarified the conclusions (P10, L10-24): "As far as the NAO impact on tracer transport is concerned, our results show that, in the recent past, NAO affected surface tracer concentrations with increased tracer concentrations over the Arctic, south Mediterranean and north Africa during positive NAO (similarly to the findings of Creilson et al. (2003), Eckhardt et al. (2003), and Christoudias et al. (2012)). Considering CO-like tracers with constant lifetime and emissions, we find that, at the end of the century, the NAO effects on pollutants will be enhanced, i.e. tracer concentrations over those areas where they are depleted during positive NAO will reduce more, while they will increase over those areas where they are transported to. This means that tracers will be transported more efficiently towards those areas which already suffer from bad air quality conditions during positive NAO, i.e. over the Arctic, south Mediterranean and Africa.

Such conclusions are confirmed also by the computation of tracer mixing ratios and transport in the Atlantic sector during high positive and low negative NAO phases. Future tracer concentrations during positive NAO will increase over Central Europe, south Mediterranean and north Africa, and reduce over north Europe and Greenland. Both the NAO amplitude changes and the NAO shift contribute to such concentration variations. For positive NAO, future tracer transport with respect to the past will get generally stronger over the North Atlantic Ocean and weaker over the Mediterranean region, enhancing the depletion of pollutants from centre-north Europe and the stagnation over south Mediterranean and north Africa. We remind that these results refer to constant emissions and idealised tracers (i.e. constant decay time)."

We hope that the repetition of the terms clarifies even further that we refer to "NAO effect on pollutants" (i.e. transport NAO-induced) of the future compared to the recent past, and not to the absolute changes in the concentrations.

**2) P4, L31:**
*"Saravan" > "Saravanan".*
Thank you for the note. We corrected the reference in the text and in the bibliography of our manuscript.

**3) P6, L1:**
*"shown in the electronic supplement" – please make explicit reference to the Figures in the supplement – e.g. in this case "Fig 1S (where S indicates supplement)". Please do the same throughout the paper.*
We named the figures in the supplement file as "Figure S1", "Figure S2", ... (we put the letter "S" in front of the number as per ACP rules) and we made explicit the references to such figures in the text. For instance, the old sentence (P6 L1): "
[revised manuscript text omitted]